# Male and Female Hormone Reading to Predict Pregnancy Percentage Using a Deep Learning Technique: A Real Case Study

**Lara Shboul [1], Kamil Fram [2], Saleh Sharaeh [1], Mohammad Alshraideh [1,*] , Nancy Shaar [1] and Njwan Alshraideh [3]**

[1] Department of Computer Science, University of Jordan, Amman 11942, Jordan
[2] Department of Obstetrics and Gynecology, University of Jordan, Jordan University Hospital, Amman 11942, Jordan
[3] Medicine School, The University of Jordan, Amman 11942, Jordan
[*] Correspondence: mshridah@ju.edu.jo

**Abstract:** Diagnosing gynecological diseases is a significant difficulty for the medical sector. Numerous patients visit gynecological clinics for pregnancies as well as for other illnesses, such as polycystic ovarian syndrome, ovarian cysts, endometritis, menopause, and others. In relation to pregnancy, patients, whether they are men, women, or both, may experience a variety of issues. As a result, in this research, we developed a proposed method that makes use of artificial neural networks (ANN) to help gynecologists predict the success rate of a pregnancy based on the reading of the pregnancy hormone ratio in the blood. The ANN was used in this test in the lab as a group of multiple perceptrons or neurons at each layer; however, in the final hidden layer, the genetic algorithm (GA) and Bat algorithm were used instead. These two algorithms are fit and appropriate for optimizing the models that are aimed to estimate or predict a value. As a result, the GA attempts to determine the testing cost using equations and the Bat algorithm attempts to determine the training cost. To improve the performance of the ANN, the GA algorithm collaborates with the Bat algorithm in a hybrid approach in the hidden layer of ANN; therefore, the pregnancy prediction result of using this method can be improved, optimized, and more accurate. Based on the flexibility of each algorithm, gynecologists can predict the success rate of a pregnancy. With the help of our methods, we were able to run experiments using data collected from 35,207 patients and reach a classification accuracy of 96.5%. These data were gathered from the Department of Obstetrics and Gynecology at the Hospital University of Jordan (HUJ). The proposed method aimed to predict the pregnancy rate of success regardless of whether the data are comprised of patients whose pregnancy hormones are in the normal range or of patients that suffer from factors favoring sterility, such as infections, malformations, and associated diseases (e.g., diabetes).

**Keywords:** pregnancy prediction; artificial neural networks (ANN); genetic algorithm (GA); Bat algorithm; machine learning





## 1. Introduction

Many patients visit gynecological clinics for pregnancy purposes, either to learn the cause of their infertility or to reassure the wellbeing of the fetus during the pregnancy period.

Regarding pregnancy, laboratory tests of men's and women's hormones in the blood that are responsible for pregnancy, such as follicle-stimulating hormone (FSH), thyroid-stimulating hormone (TSH), luteinizing hormone (LH), Estradiol (E2), prolactin, anti-Mullerian hormone (AMH), and seminal fluid analysis, can help identify the problems the couple is experiencing with pregnancy [1]. The effect of the seminal fluid analysis in the success rate of pregnancy is shown in Table 1.

**Table 1.** The effect of the seminal fluid analysis in the success rate of pregnancy [2].

| | |
|---|---|
| Count | If the count is less than 15 million/ejaculate: the chance of pregnancy is 11.3%<br>If the count is more than 39 million/ejaculate: the chance of pregnancy is 22.6% |
| Volume | If the volume is less than 2 mL: the chance of pregnancy is 5.6%<br>If the volume is more than 2 mL: the chance of pregnancy is 11.2% |
| Viscosity | In normal viscosity [15 to 30 min], the chance of pregnancy is about 18%,<br>But with hyper viscosity [ranging from +1, +2, +3, +4] is ranging from less than 6% to less than 12%. |
| Motility | More than 20 million motility is associated with 18%,<br>Less than 5 million motility is associated with less than 1%. |
| Concentration; The sperm concentration (no./mL semen) | If the number is less than 15 million/mL, the chance is less than 8%.<br>If the number is more than 40 million/mL, the chance is less than 16%. |

Relating to the pregnancy success rate and the hormones that cause pregnancy, such as FSH, estrogen, LH, etc.:

- High levels of (FSH) in a woman's blood could reduce the chances of getting pregnant by 75%;
- Abnormal levels of estradiol, an important form of estrogen, decrease the chances of pregnancy by 65%;
- Insufficient levels of (LH), which prompts the ovaries to release an egg and begin generating progesterone, can also account for 25% of reproductive issues [3].

Numerous studies have been conducted on minor and specific diagnostic challenges in the field of medical diagnosis. Typically, the proper diagnosis is documented in medical records from specialized institutes or departments. All that is required to start a learning process in computer software is the entry of patient data with accurate diagnoses that are known [4].

An artificial neural network (ANN) is a computing paradigm that was inspired by the biological neural system and belongs to the computational intelligence family [5]. Researchers have offered a number of strategies and methodologies to optimize issues.

The Bat method is prone to local optima, and due to its limited capacity for global exploration, its optimization findings are insecure [6]. To address these issues, our research proposes a ground-breaking Bat algorithm built on an integration approach (IBA). The integration approach guarantees that the global search capability is maintained by adaptively selecting an appropriate operator to execute a global search.

The IBA disrupts the local optimum by utilizing a linear combination of Gaussian functions with varied variances to prevent becoming trapped in local optima [7].

Genetic algorithm is a search heuristic-based algorithm, based on Charles Darwin's natural selection hypothesis. This algorithm mimics natural selection, where the fittest individuals are selected for reproduction in order to produce the offspring of the next generation [8].

We were able to obtain the most precise results in such sensitive models by utilizing the genetic and Bat algorithms to enhance the results of the prediction models. We eventually reached this proposed method, which incorporates a hybrid approach to enhance the accuracy result of the ANN, after training the models using several techniques. This study's primary contributions can be summed up as follows:

- To receive an accurate prediction result;
- Boost the neural network's result;
- Understand how to combine two distinct algorithms in the ANN's hidden layer.

This paper's remaining sections are organized as follows: Section 2 describes the research that is associated with this research. Section 3 provides descriptions and explanations of the methodology and the process used in this research. The results are

discussed, compared, and their performance is evaluated in Section 4. In Section 5, the essay is concluded.

## 2. Literature Review

Fossil fuels, which are used in traditional power generation methods, are a major cause of global environmental problems like global warming and climate change. Battery energy storage systems (BESSs) are frequently used to store electrical energy for backup, balance power demand and generation during peak hours, and promote energy efficiency in a pollution-free environment. As a result, renewable energy sources (RESs) are employed for power generation. In order to ensure battery safety, reduce maintenance costs, and lessen BESS discrepancies, accurate battery state of health (SOH) prediction is essential. The accurate forecasting of power use is essential for avoiding power shortages and excess, and it is impossible to directly acquire the complex physicochemical characteristics of battery deterioration. So, for multi-step SOH and power consumption forecasting, a unique hybrid architecture named "CL-Net" based on convolutional long short-term memory (ConvLSTM) and long short-term memory (LSTM) is suggested in this study. First, raw data pertaining to battery SOH and power consumption are gathered and subjected to preprocessing for data purification. After processing the input, ConvLSTM layers receive it and extract spatiotemporal features to create their encoded maps. The decoded features are then passed to fully linked layers for the final multi-step forecasting using LSTM layers. Finally, using three different time series datasets—the national aeronautics and space administration (NASA) battery, individual household electric power consumption (IHEPC), and domestic energy management system—a thorough ablation study is carried out on various combinations of sequential learning models (domestic energy management system (DEMS)). On the NASA battery and IHEPC datasets, the suggested CL-Net architecture reduces root mean squared error (RMSE) up to 0.13 and 0.0052, respectively, in comparison to the state-of-the-art architecture.

These experimental findings demonstrate that, when compared to the state-of-the-art architecture, the suggested architecture can provide reliable and accurate SOH and power consumption forecasts [8].

A persistently high blood glucose level, often known as blood sugar, raises the chance of developing the condition diabetes. It is the main energy source required by the organism for proper operation. Machine learning algorithms are creating intelligent systems that are reshaping every aspect of our life, including the healthcare industry. An artificial neural network prediction model was created in this study to identify diabetes mellitus.

The dataset was gathered from the Kurdistan area, which interviewed both diabetic and non-diabetic pregnant mothers. The goal of this study was to use a neural network model architecture to train neural networks while minimizing the error function. Based on the ANN model's design procedure, the neural network's error rate dropped during training after it was created and the prediction's accuracy rose to 91%. Based on the findings, it can be concluded that the ANN algorithm enhances prediction accuracy [9].

A digital realm called Metaverse (MS) is reachable through a virtual setting. It is created by fusing virtually enhanced physical reality with digital reality. For students in learning and educational contexts, Metaverse (MS) delivers improved immersive experiences and a more engaging learning experience. It is a setting for enlarged and synchronous conversation where various users can exchange experiences. The purpose of the current study is to assess how students perceive MS's use in the United Arab Emirates (UAE) for medical and educational purposes. A total of 1858 college students were polled as part of this study to assess this model.

The adoption constructs included in the conceptual framework of the study were the technology acceptance model (TAM), personal innovativeness (PI), perceived compatibility (PCO), user satisfaction (US), perceived triability (PTR), and perceived observability (POB) factor, the model used in the study, which connected both individual- and technology-based factors, making it distinctive. Additionally, hybrid analyses like machine learning (ML)

methods and structural equation modeling were employed in the study (SEM). In order to evaluate the importance and performance of the elements, the current study also uses importance-performance map analysis (IPMA). The study reveals that the US is a crucial factor in predicting users' inclination to use the Metaverse (UMS). The results of the current study are beneficial for stakeholders in the educational sector because they will help them understand the significance of each component and help them make strategies based on the relative weights of the various factors. The study also methodologically adds to the literature on information systems (IS) because it is one of the few studies to use ML algorithms, a complementing multi-analytical technique [10].

For more than 50 years, predicting pregnancy has been a major issue in women's health. Previous datasets were gathered through highly selected medical research but the current rise of women's health tracking smartphone applications opens up the possibility of reaching a far larger audience. The practicality of forecasting pregnancy with mobile health tracking data, on the other hand, is unclear. So, researchers proposed to use data from a women's health tracking app (Clue by BioWink GmbH) to build four models—a logistic regression model and three LSTM models—to estimate a woman's likelihood of becoming pregnant.

Researchers evaluated the models on a dataset that contains 79 million logs from 65,276 women with ground truth pregnancy test data. The results showed that their predicted pregnancy probabilities meaningfully classified women: women in the top 10% of predicted probabilities have an 89% chance of becoming pregnant over six menstrual cycles, compared to a 27% chance for women in the bottom 10%. They devised a method for deriving interpretable temporal trends from their deep learning models, which they demonstrate are compatible with past fertility studies. The results showed how women's health tracking data may be used to forecast pregnancy in a larger population [11].

In 2020, there was a model developed for gynecological disease diagnosis. The gynecological diseases were polyps, infection, fibroids, prolapse, cancer, endometrial hyperplasia, migrants, amenorrhea, abortion, dysmenorrhea, and infertility. The proposed system was developed using one of the most widespread machine learning techniques, a feed-forward neural network that was trained using the Rprop training algorithm [12].

It consisted of an input layer with 54 neurons (representing the input variables of each patient such as age, marital status, pregnancy, etc.), with four hidden layers where the number of neurons in the first, second, third, and fourth hidden layers was 27, 14, and 7, and 3 respectively and an output layer that produced the type of the disease from which the patient suffered. Moreover, ten-fold cross-validation was used to access the generalization of the proposed system using 550 patients' medical records collected from the gynecology clinics at JUH, which were obtained for the polyps, infection, fibroids, prolapse, cancer, endometrial hyperplasia, migrants, amenorrhea, abortion, dysmenorrhea, and infertility, producing an average of 94.4% classification accuracy [12].

The current study is based on an integrated research model that was created by fusing a technology acceptance model (TAM) dimensions with other aspects of wristwatch effectiveness, like content richness and user pleasure (SAT). TAM is used to identify variables affecting the uptake adoption of the smartwatch (ASW). The current study, which facilitates and enhances the effective participation of doctors and patients, focused on aspects that affect smartwatch acceptability and use in the medical field. The conceptual framework of the current study investigates the close relationship between the notions of user satisfaction and content richness and the two-term TAM variables of perceived ease of use (PEU) and perceived usefulness (PU). The flow theory (EXP) is also used to calculate the smartwatch's efficiency. The study also evaluates ASW involvement and control using the flow theory. A sample of 489 medical professionals, including doctors, nurses, and patients, participated in the study. The study used a hybrid analysis approach that combines structural equation modeling (SEM) with a deep learning-based artificial neural network (ANN).

The significance and effectiveness of the factors impacting ASW were also determined by the study using importance-performance map analysis (IPMA). User happiness is the

most important indicator of intent to use a medical wristwatch, according to the ANN and IPMA analyses. The structural equation model applied to the sample reveals that the variables SAT, PU, PEU, and EXP have a substantial impact on respondents' intentions to use a medical wristwatch [13].

### 3. Methodology

The first stage in the proposed method was to collect all features that are significantly influencing the estimation of pregnancy in order to collect the raw dataset. The next step involved the proposed method using a feature selection process to select a representative set of attributes from the set of original attributes. In this phase, the proposed method used the correlation-based feature selection, principal component analysis, information gain ratio-based feature selection, and minimum redundancy maximum relevance methods. The aim of This level was to build a representative set to keep only the relevant and important attributes from the raw dataset.

In the next phase, the proposed method used an artificial neural networks (ANN) for training the network, using a hidden layer of neurons. The last layer of the suggested strategy employs a hybrid of the genetic algorithms (GA) and Bat algorithms that were employed to calculate the cost of training and testing. Figure 1 represents the architecture of the proposed algorithm.

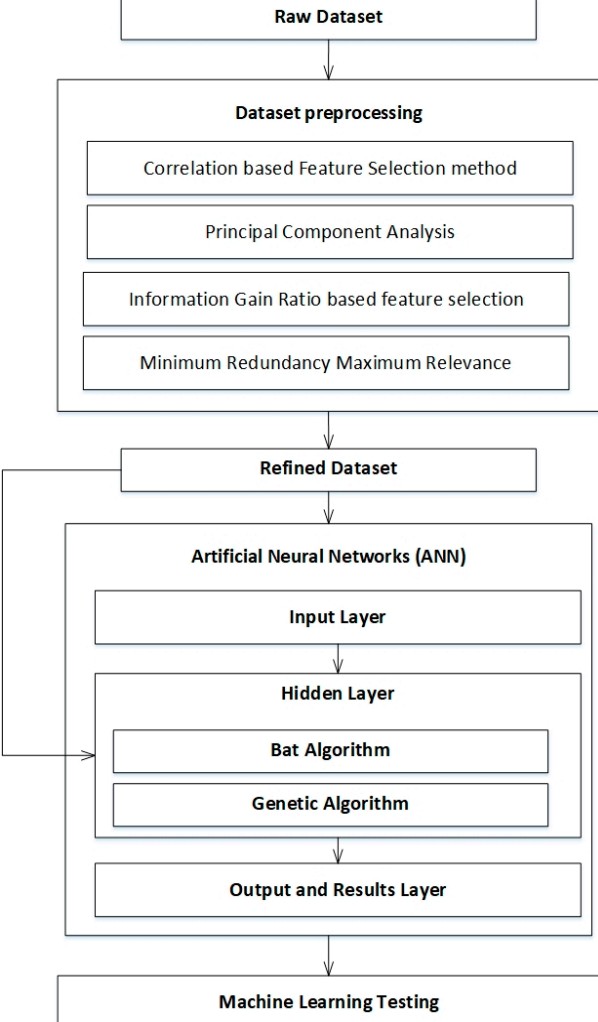

**Figure 1.** General framework of the proposed method.

### 3.1. Dataset

The collected dataset (raw dataset) used in this study, which has over 35,207 records, was created using authentic patient records for various cases (wives and husbands) treated at Jordan University Hospital (JUH). Each record contains different readings for each woman (current age, FMC, FSH, TSH, E2, LH, and AMH) and man (LH, FSH, TSH, and seminal fluid analysis parameters that shows count, motility, volume, viscosity, and concentration).

This study uses two sets of data: the first focuses on male pregnancy hormones while the second focuses on female pregnancy hormones and their combined effects.

Table 2 contains the percentage of pregnancy hormones, which aids in the decision-making process and the total was 185% in real percentage. This value is not accepted in the proposed method because these kinds of values are random; therefore, a mathematical equation was used in this case to make this value readable for the system out of 100%.

**Table 2.** The hormone's effects as a percentage in real percent and system percent.

| Female | Real Percent | System Percent |
| :---: | :---: | :---: |
| FSH | 75% | 0.41 |
| TSH | 20% | 0.11 |
| E2 | 35% | 0.19 |
| LH | 25% | 0.14 |
| AMH | 30% | 0.16 |
| Total | 1.85 | 100% |

### 3.2. Dataset Preprocessing

Dataset preprocessing is a set of techniques that aim to simplify the dataset by eliminating some of the non-descriptive and unnecessary attributes from the original dataset [14]. In this study, the proposed method makes use of a variety of techniques and algorithms to select the relevant features from the raw dataset, which leads to earning more facilities for data interpretation and visualization. When the first method has completed its task, it passes the outcome to the second one so that it can finish independently preprocessing the data. Table 3 describes the used feature selection methods in dataset preprocessing.

**Table 3.** The feature selection methods in dataset preprocessing.

| Method | Description |
| :---: | :---: |
| Correlation-based feature selection method (CFS) | Aims to have new subsets of features highly correlated with a specific class (classes), and uncorrelated to each other (attributes). Is used for finding the association between the continuous feature and class feature |
| Principal component analysis (PCA) | Is a technique for reducing the dimensionality of such dataset increasing interpretability but at the same time minimizing information loss? The proposed method, this technique aims to identify all uncorrelated features. |
| Information gain ratio-based feature selection (IGR) | Splitting the attributes pattern distribution into classes, where a gain ratio of attributes decreases as the value of split information increases |
| Minimum redundancy maximum relevance | punish a feature's relevance based on its redundancy. |

### 3.3. Artificial Neural Networks (ANN)

The artificial neural network (ANN) was used as a group of multiple perceptrons/ or neurons at each layer. Because inputs are solely processed in the forward direction, the ANN is also known as a feed-forward neural network. The ANN contains three main layers (input, hidden, and output) [5]. The input layer accepts the inputs and is used to read the dataset, while the hidden layer processes the inputs. We include our contribution by adding the hybrid layer of GA and Bat algorithm in the network's last layer.

The hybrid layer is also used as the output layer because it produces the result.

*3.4. Hybrid of GA and Bat Algorithm*

The Bat algorithm (BA) is a new meta-heuristic optimization algorithm derived by simulating the echolocation system of bats [15]. The basic idea of the BA is to imitate the echolocation behavior of bats with a local search of bat individuals for achieving the global optimum. It is developing into a promising solution because of its success in addressing a variety of problems and in applying the algorithm to an optimization problem. A "bat" generally, represents an individual in a population. The environment in which the artificial bat lives is mainly the solution space and the states of other artificial bats. Its next movement depends on its current state, velocity, and environmental state (including the quality and state of the best bats in the swarm).

This phase of applying the hyper-approach is divided into three main levels. In the first level, the proposed method works to find the training cost using the Bat algorithm, while the second level aims to find the testing cost, and the final phase is used as an output layer for the ANN algorithm and returns the pregnancy prediction results using the GA algorithm.

In the proposed method, the genetic algorithm (GA) works to find the testing cost; GA has several steps starting with the initialization phase and finishing with the termination phase [16]. The GA algorithm works with the Bat algorithm as a hybrid approach to increase the performance of the ANN algorithm and optimize the pregnancy prediction based on the flexibility of each algorithm.

After implementing the proposed Bat and GA algorithm methods, the system analysis was carried out. In the first phase of measuring the results, the cost of each training dataset (training cost) and testing dataset (testing cost) is calculated, then the difference between these values of cost is taken into consideration (C). On the other hand, the system shows the scales (D) of values achieved in the normal cases for each woman and man.

After that, we calculate $R1$ and $R2$ according to the formulas below:

$$R1 = testingCost * D \tag{1}$$

$$R2 = C - R1 \tag{2}$$

According to the threshold value 500.00, the $R2$ value is the final result that determines whether a pregnancy is possible or not. Then, we find the actual results (the real medical results), classified results (the proposed method results), and final results (matching results between the actual and classified results).

The cross-validation technique, which is an evaluation of a predictive model's accuracy, was then used to divide the obtained data into subsets. It is calculated by dividing the total number of correctly classified cases in the dataset by the total number of instances. Partitioning a sample of data into subsets, analysis on one subset (referred to as the training set), and testing the analysis on the other subset constitutes one round of cross-validation (called the testing set). The ANN experiences overfitting. To prevent overfitting, a third subset known as the validation set is used. Additionally, several cross-validation rounds are carried out using various divisions to reduce variability and prevent bad splits that could lead to overfitting, and the tested results are averaged over the rounds.

Ten-fold cross-validation was employed in this study to examine the network model's generalizability. To represent the training and validation, a data split of 70% and 30% of the total samples was used.

## 4. Experiments

This section contains the experiment results for the proposed methods, where the artificial neural network (ANN) is used as a group of multiple perceptron or neurons at each layer according to the mean square error. To perform the experiments, we used six experiments according to the number of hidden layers (5, 10, 15, and 20 layers) for the

ANN. All the experiment results are figured out from MATLAB program. Figure 2 shows the ANN networks used for the training step:

**Figure 2.** The training for the ANN network.

To compare the results and identify the optimal layers based on prior measurements and the required run time, we repeated the training for each of the 5, 10, 15, 20, 25, and 50 layers.

An epoch is one cycle through the entire training dataset in artificial neural networks, shown in Figure 3. Usually, a neural network requires more than a few epochs to train (6 epochs for training 20 hidden layers). In other words, if a neural network receives training data from the system for multiple epochs in various patterns, it produce a better generalization when exposed to new, "unseen" input (test data). An epoch is often mixed up with iteration. Iteration is the number of batches or steps for the training data's partitioned packets that are required to complete one epoch. Figure 3 shows the training states for all required epochs.

When we categorize data, we primarily group them into those categories without considering what the data might indicate. When creating a histogram network that aims to create a chart that provides the network with some extremely useful information about how data are distributed, basic sorting into categories like male or female or yes or no does exist in statistics. As a result, the network intends to cautiously select its categories or classes. In these cases, bins refer to physical bins into which objects may be sorted, as seen in Figure 4.

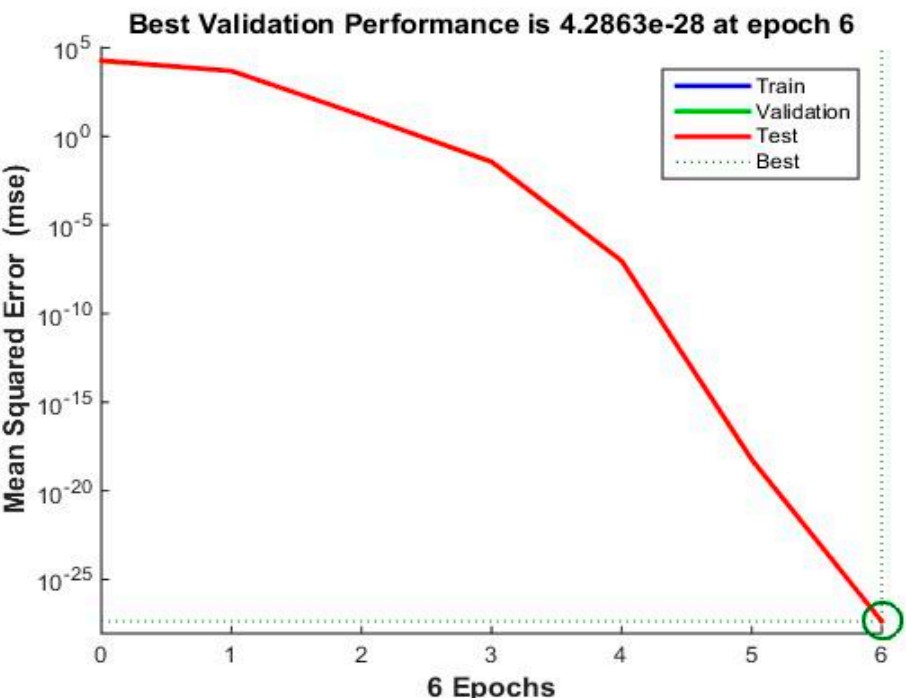

**Figure 3.** Performance of 50 layers.

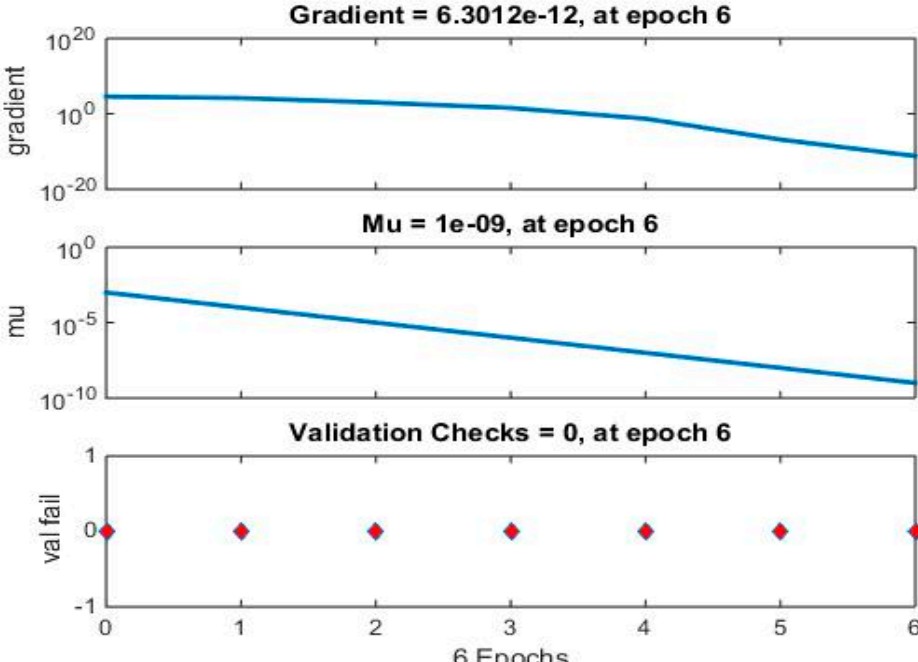

**Figure 4.** Training state of ANN model.

Regression ANNs predict an output variable as a function of the inputs. The input features (independent variables) can be categorical or numeric types. However, for regression ANNs, we require a numeric dependent variable. If the output variable is a categorical variable (or binary) the ANN functions as a classifier. Figure 5 shows the regression for the ANN network.

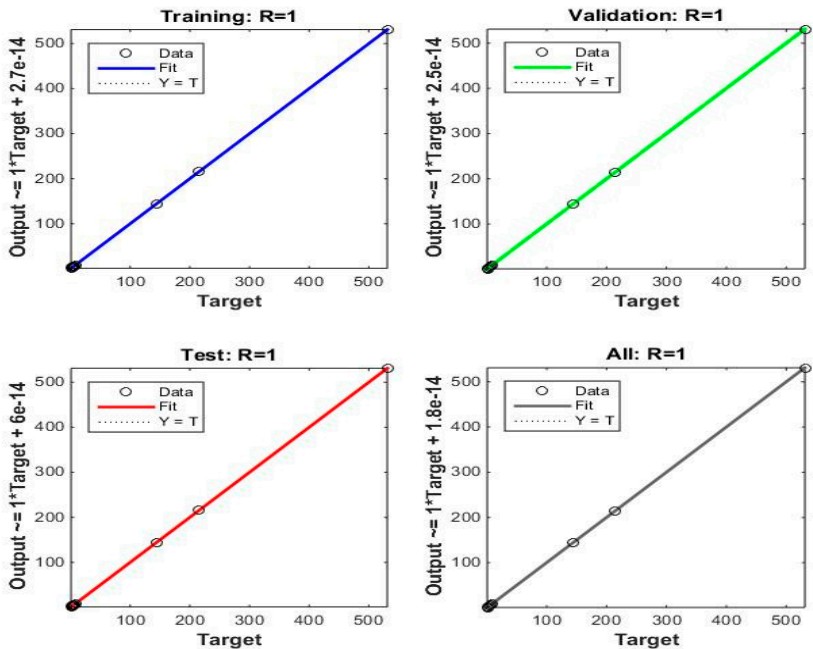

**Figure 5.** Linear regression for ANN training, validation, testing, and combined input datasets.

*4.1. Simulation Results*

First, all experiments are divided into groups (A, B, C, D, and E) based on the age of the women, and then 100 instances are measured (20 instances for each class). The experiment with 100 cases (wives and husbands) demonstrates the best features that directly affect pregnancy prediction in the proposed approach. Age, FMC, FSH, TSH, E2, LH, and AMH are general features of women, whereas LH, FSH, TSH, volume, count, viscosity, motility, and concentration are general features of men.

*4.2. ANN Performance*

One of the unsolved problems in this field of research is how to optimize the number of hidden layers and neurons in each hidden layer for a feed-forward neural network. As a result, a number of academics have put forth some general guidelines for figuring out the ideal number of hidden neurons for a given application, depending on the volume of training data and the difficulty of the classification issue to be solved [17].

The best number of hidden layers and units depends primarily and complexly on the number of input and output units, the number of training cases, and the difficulty of the classification problem to be learned, as demonstrated by Kavzoglu, as cited in [18]. In most cases, there is no way to determine the best number of hidden layers without training multiple networks and estimating the generalization error of each [19].

This allowed us to decide the ideal number of hidden layers and neurons for each hidden layer using an iterative procedure. The best neural model with the highest classification accuracy is selected iteratively using a ten-fold cross-validation technique, and the process goes like this:

Step 1: The formula in equation is used to determine the number of neurons in the first hidden layer when evaluating the network architecture with just one hidden layer.

$$n_f = (n_i + n_o)/2$$

where $n_i$ is the number of input layer neurons, $n_o$ is the number of output layer neurons, and $n_f$ is the number of neurons in the first hidden layer. Apply the ceiling and floor operations if $nf$ is not a fixed number to obtain two values. For more accurate results, add a third number, floor($n_f$) − 1, producing three numbers of neurons in the first layer to begin with.

Step 2: Add a second layer, which has half as many neurons as the first layer.

Step 3: Carry out Step 2 repeatedly until there is just one neuron in the layer.

To find the classification accuracy for a model, we must first calculate the classification accuracy for each fold and, after that, average the results over the ten folds.

The performance of the ANN algorithm is shown in this section based on the run time required and the error rate. Figure 6 shows the average run time for each experiment (5, 10, 15, 20, 25 and 50 layers):

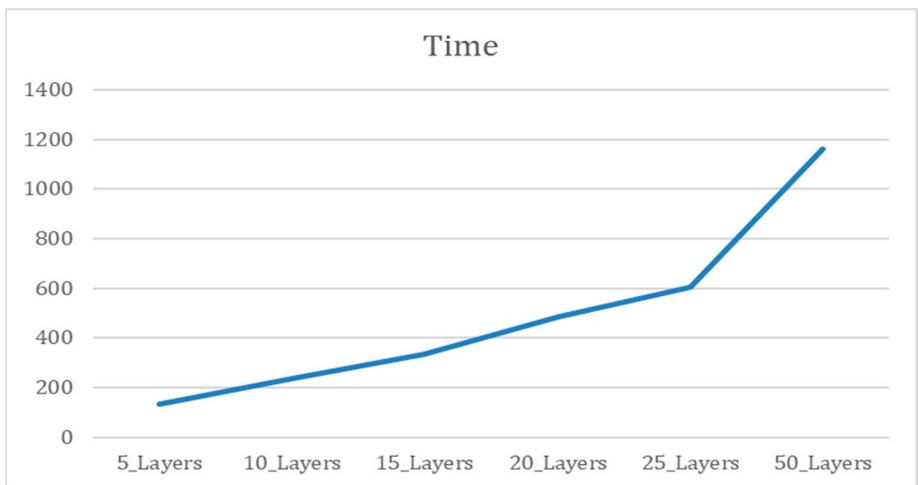

**Figure 6.** The average run time for each experiment.

According to Figure 6, the ANN requires 131.3239104 s as a total average for instances in the first simulation run when it is conducted with five hidden layers. With 10 hidden layers, the ANN needs 235.4020 s as a total average for instances in the second experiment. However, for 15 hidden layers, the ANN requires 336.4227 s as a total average for instances in the third simulation run. The ANN requires 486.7220 s as a total average for instances in the fourth simulation run when employing 20 hidden layers. However, when using 25 hidden layers, the ANN requires 604.8766 s as a total average for instances in the fourth simulation run. The ANN requires 1160.7525 s as a total average for instances in the final simulation run with 50 hidden layers.

To identify the optimal simulation run according to the run time, Figure 6 displays the simulation run when calculating the run time for 5 layers.

According to Figure 7, we can observe the preference for the third simulation run when using 15 hidden layers, where the average run time for every five layers needs 112.1408919 s. On the other hand, the worst case of the simulation run was for the first simulation run with only five layers, where the average run time reached 131.3239 s.

### 4.3. Bat Algorithm and Genetic Algorithm

This section presents the final results for the proposed method following the system analyses utilizing both the Bat and GA algorithms. In order to measure the results, we first calculated the cost for each training dataset (training cost) and testing dataset (testing cost). We then took the difference between these cost values (C). Nevertheless, the system displays the scales (D) of values achieved in the normal cases for each woman and man.

After that, we calculate R1 and R2 according to the formulas below:

According to the threshold value (500.00), the R2 value is the final result that determines whether or not a pregnancy is likely. Then, we come across the actual results (i.e., the real medical results), the classified results (i.e., the proposed method results), and the final results (i.e., the matching results between the actual and classified results). Table 4 shows the classified results for all classes.

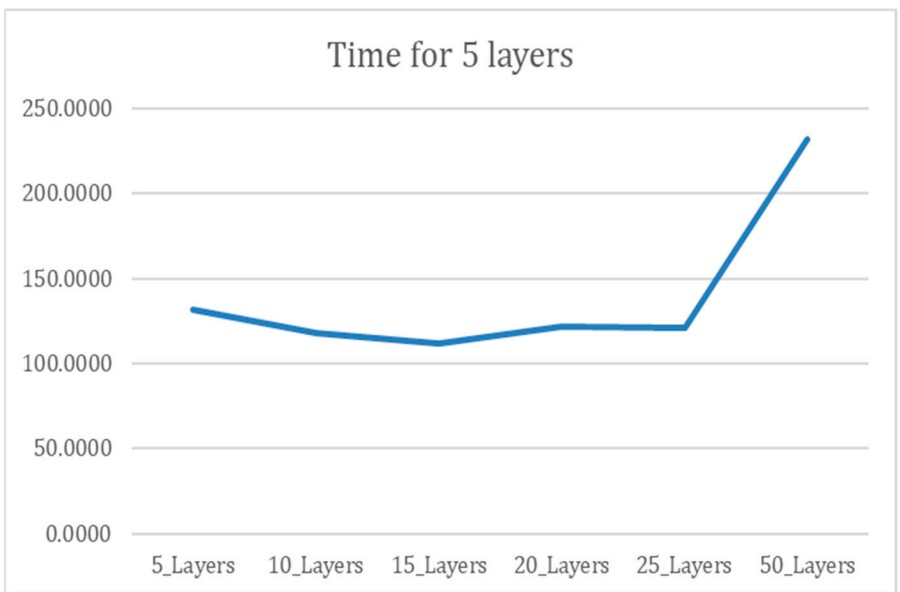

**Figure 7.** The average run time for every 5 layers in all simulation runs.

**Table 4.** Classified results for all classes.

| Class | Correct | Wrong | Accuracy | Error Rate |
|---|---|---|---|---|
| A | 5 | 0 | 100% | 0 |
| B | 5 | 0 | 100% | 0 |
| C | 5 | 0 | 100% | 0 |
| D | 4 | 1 | 85% | 15% |
| E | 4 | 1 | 95% | 5% |
| Total | 23 | 2 | | |

Table 4 shows the variation in results between the classes A, B, and C, each achieving a high accuracy of prediction (100%) for each of the true positive and true negative values, and the lowest possible error rate (0%). Class D accuracy was 85% accurate with a 15% error rate. While class E accuracy was 95% with a 5% error rate.

*4.4. Data Analysis and Interpretation*

We utilized the KNIME Analytics Platform for data analysis and interpretation, which enabled us to test the accuracy of the system's MATLAB results. Before starting with KNIME, we first converted the Excel file's data results to a comma-separated values (CSV) file in order to make them compatible with the KNIME model. The KNIME model generates a confusion matrix containing each of the following entries: correct classified, wrong classified, accuracy, and error. Figure 8 shows the KNIME model used for building the confusion matrix.

Support vector machine (SVM), random forest, decision tree, simple regression tree, and multilayer perceptron (MLP) are the five classifiers and measures included in the KNIME Analytics Platform's proposed model, as shown in Figure 8. These classifiers tested the performance and accuracy of the proposed method. Using a scorer node, each classifier generates a confusion matrix file and each matrix is dependent on a separate set of measures, as indicated in the Table 5.

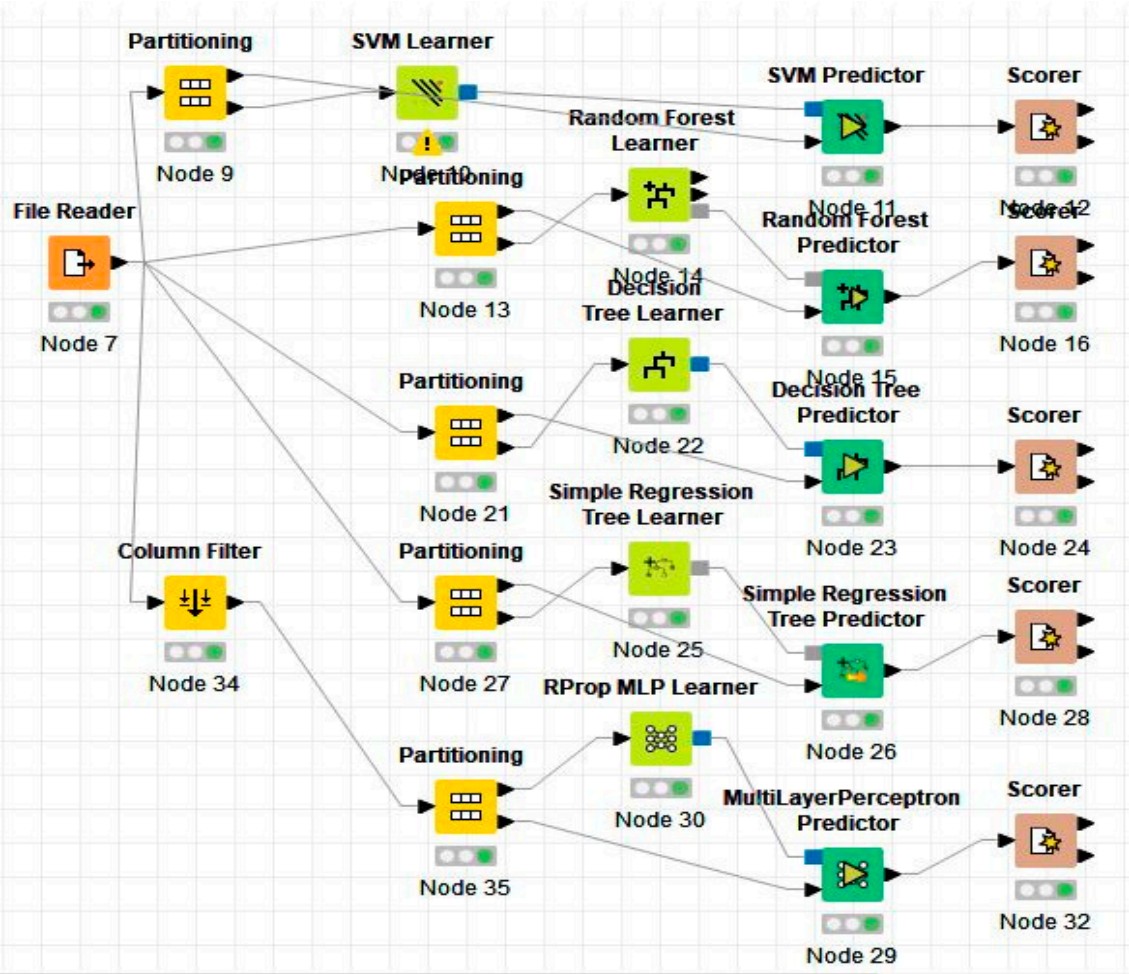

**Figure 8.** KNIME Analysis.

**Table 5.** Testing measures.

| Measures | SVM | | Random Forest | | Decision Tree | | Simple Regression Tree | |
|---|---|---|---|---|---|---|---|---|
| | **F** | **T** | **F** | **T** | **F** | **T** | **F** | **T** |
| True-Positives | 0 | 66 | 3 | 66 | 3 | 65 | 30 | 35 |
| False-Positives | 4 | 0 | 1 | 0 | 0 | 2 | 1 | 4 |
| True-Negatives | 66 | 0 | 66 | 3 | 65 | 3 | 35 | 30 |
| False-Negatives | 0 | 4 | 0 | 1 | 2 | 0 | 4 | 1 |
| Recall | - | 0.943 | 1 | 0.985 | 0.6 | 1 | 0.882 | 0.972 |
| Precision | 0 | 1 | 0.75 | 1 | 1 | 0.97 | 0.968 | 0.897 |
| Sensitivity | - | 0.943 | 1 | 0.985 | 0.6 | 1 | 0.882 | 0.972 |
| Specificity | 0.943 | - | 0.985 | 1 | 1 | 0.6 | 0.972 | 0.882 |
| F-measure | - | 0.971 | 0.857 | 0.992 | 0.75 | 0.985 | 0.923 | 0.933 |

Table 5 shows the confusion matrix for the KNIME classifiers that were calculated according to the previous measures in Table 6.

**Table 6.** Confusion matrix for KNIME classifiers.

| Classifier | Correct Classified | Wrong Classified | Accuracy | Error |
|---|---|---|---|---|
| SVM | 66 | 4 | 94.286% | 5.714% |
| Random Forest | 69 | 1 | 98.571% | 1.429% |
| Decision Tree | 68 | 2 | 97.143% | 2.857% |
| Simple Regression Tree | 65 | 5 | 92.857% | 7.143% |
| MLP | 64 | 3 | 90.00% | 10.00% |

Table 6 shows that the results for the five classifiers are similar to the final results for the proposed method displayed in Table 5, especially with the SVM classifier, whereas the SVM's accuracy was 94.286% and its error rate was 5.714%. In comparison to other classifiers, the random forest classifier delivered the best results, with an accuracy rate of 98.571% and an error rate of 1.429%. Conversely, the accuracy of the decision tree classifiers was 97.143% and its error rate was 2.857%. However, when compared to other classifiers, the simple regression tree classifier had the lowest results, with an accuracy of 92.857% and an error rate of 7.143%. Finally, a multilayer perceptron (MLP) achieved an accuracy of 90% and an error rate of 10%.

Twenty hidden layers were added to the models after we started with just one. We can see that the best architecture, which has the dimensions 54-27-14-7-3-1 and a classification accuracy of 96.5%, was created utilizing four hidden layers. A single hidden layer did not produce satisfactory results but when a second hidden layer was added, the best classification accuracy increased from 73.6% to 88%, and when three hidden layers were added, it increased to 93.6%. The highest accuracy was achieved with a percentage of 96.5% by adding a fifth layer. Six and above hidden layers did not improve the outcomes and actually decreased accuracy.

This led us to the conclusion that the only method suitable to choose the ideal neural network architecture for a given task is testing.

As can be seen, fold nine had the best classification accuracy, with a 100% accuracy rate, while fold seven had the lowest accuracy, at 80%. Additionally, the overall categorization accuracy was 96.5%. The way, that the data divided for each fold had an impact on these outcomes. Since the data pattern in the test set was highly similar to examples in the training sets, the fold one, which achieved 100% accuracy, might have had high classification accuracy.

Table 7 shows the confusion matrix of the diagnostic result of 10-fold cross-validation that proves that the tuning of the parameter does not affect the accuracy of the methods.

**Table 7.** Confusion matrix of the diagnostic result of the pregnancy hormones: 10-fold cross-validation.

| Disease Type | Count | Volume | Viscosity | Motility | Concentration | FSH | TS | LH | E2 | AMH | Estrogen | Row Sum |
|---|---|---|---|---|---|---|---|---|---|---|---|---|
| Count | 45 | 2 | 0 | 1 | 2 | 1 | 0 | 0 | 0 | 0 | 0 | 50 |
| Volume | 0 | 45 | 1 | 0 | 0 | 3 | 2 | 0 | 0 | 0 | 0 | 50 |
| Viscosity | 1 | 0 | 47 | 1 | 1 | 0 | 0 | 0 | 0 | 0 | 0 | 50 |
| Motility | 0 | 0 | 0 | 49 | 1 | 0 | 0 | 0 | 0 | 0 | 0 | 50 |
| Concentration | 0 | 0 | 0 | 0 | 49 | 1 | 0 | 0 | 0 | 0 | 0 | 50 |
| FSH | 0 | 0 | 1 | 0 | 0 | 49 | 0 | 0 | 0 | 0 | 0 | 50 |
| TSH | 0 | 0 | 0 | 0 | 0 | 1 | 49 | 0 | 0 | 0 | 0 | 50 |
| LH | 0 | 0 | 0 | 0 | 0 | 0 | 0 | 50 | 0 | 0 | 0 | 50 |
| E2 | 0 | 0 | 0 | 0 | 0 | 0 | 0 | 0 | 50 | 0 | 0 | 50 |
| AMH | 0 | 0 | 0 | 0 | 0 | 0 | 0 | 0 | 1 | 49 | 0 | 50 |
| estrogen | 0 | 0 | 0 | 0 | 0 | 0 | 0 | 0 | 0 | 1 | 49 | 50 |
| column sum | 46 | 47 | 49 | 51 | 53 | 55 | 51 | 50 | 51 | 50 | 49 | 550 |

The confusion matrix in Table 7 can be used to determine the accuracy for each class. Table 8 displays the percentage of accurate identification of the success rate of pregnancy by the ANN.

**Table 8.** Proportion of correctly classified hormones.

| Pregnancy Hormones | Accuracy |
|:---:|:---:|
| Count | 95% |
| Volume | 97% |
| Viscosity | 94% |
| Motility | 96% |
| Concentration | 96% |
| FSH | 95% |
| TSH | 95% |
| LH | 100% |
| E2 | 98% |
| AMH | 98% |
| estrogen | 98% |
| The average | 96.5% |

*4.5. Comparing Results*

Next, we compared the results for the proposed approach with Liu et al. (2019), where they used several scenarios for testing their proposed method for the logistic regression, LSTM, LSTM with BMS, and LSTM with the user embeddings. Table 9 shows the collecting models and classifiers for all previous methods with the proposed method's results for our study.

**Table 9.** The comparison of the proposed method's results with the related work.

| Study | Classifier | Accuracy |
|:---:|:---:|:---:|
| Liu et al. (2019) | Logistic regression | 63.0% |
| Liu et al. (2019) | LSTM | 65.0% |
| Liu et al. (2019) | LSTM + BMS | 64.0% |
| Liu et al. (2019) | LSTM + user embedding's | 67.0% |
| Proposed method | SVM | 94.2% |
| Proposed method | Random Forest | 98.5% |
| Proposed method | Decision Tree | 97.1% |
| Proposed method | Simple Regression Tree | 92.8% |
| Proposed method | A multilayer perceptron (MLP) | 90.00% |

In general, we can notice the preference of the results for the proposed method compared to Liu et al. (2019), where the accuracy for the SVM classifier reached 94.2%, 98.5% for random forest, 97.1% for the decision tree, 92.8% for the simple regression tree, and 90.0% for the multilayer perceptron (MLP).

*4.6. Algorithms and Time Complexity*

We concentrated on prediction and its problems as well as the model performance after it completes the entire assignment. The model may be evaluated based on several elements and factors. Time is the primary determinant of performance and cost; every successful system should complete tasks correctly while utilizing system resources in the best and most efficient manner. As a result, this section displays the proposed system algorithms as well as the time complexity of each algorithm.

In this part, we assess the key operations and algorithms and determine their time complexity as follows:

This phase is divided into three main levels. In the first level, the proposed method works to find the training cost using the Bat algorithm, while the second level aims to find

the testing cost, and the final phase is used as an output layer for the ANN algorithm and returns the pregnancy prediction results using the GA algorithm.

### 4.6.1. The Time Complexity for the Bat Algorithm

The suggested method's computing complexity is determined by a variety of factors, including the number of pulse rates, iterations, and tasks. The Bat algorithm, which sorts universes on each iteration and has a complexity of $O(n \log(n))$ in the best case and $O(n 2)$ on the average and worst cases, are used for the sorting mechanism. Every variable in the universe is selected using a roulette wheel, and since our implementation of the roulette wheel selection mechanism has a complexity of $O(\log(n))$, the suggested model scheduling algorithms have the following total complexity times: $O (i * n 2 + n * t * \log(n))$. Figure 9 shows simple code of the Bat algorithm.

```
% Start the iterations -- Bat Algorithm
for i_ter=1:N_iter,
        % Loop over all bats/solutions
        for i=1:n,
          Q(i)=Qmin+(Qmin-Qmax)*rand;
          v(i,:)=v(i,:)+(Sol(i,:)-best)*Q(i);
          S(i,:)=Sol(i,:)+v(i,:);
          % Pulse rate
          if rand>r
              S(i,:)=best+0.01*randn(1,d);
          end

      % Evaluate new solutions
          Fnew=Fun(S(i,:));
      % If the solution improves or not too loudness
          if (Fnew<=Fitness(i)) & (rand<A) ,
              Sol(i,:)=S(i,:);
              Fitness(i)=Fnew;
          end

          % Update the current best
          if Fnew<=fmin,
              best=S(i,:);
              fmin=Fnew;
          end
        end

end
% End of the main bat algorithm and output/display can be added here.
```

**Figure 9.** A Bat algorithm code sample.

### 4.6.2. The Time Complexity for the Genetic Algorithm

You can also observe that the genetic algorithm goes through several iterations. At first, a set of solutions S are produced at random (S is called a population). The prices of S solutions are calculated. In each cycle, some operations are carried out on the S solutions, including crossover and mutation. The top k solutions in k are retained in S, and we carry on as before. We output the top answer we discovered after the final iteration.

You can see that an iteration's time cost depends on its internal operations, which are frequently straightforward to implement and are problem-specific (e.g., crossovers, mutation, and others, finding the best k distinct solutions, generating random solutions, calculating the cost of solutions for S, etc.). They typically depend on the volume of a solution.

This algorithm is used to determine the testing cost and calculate it; it has an $O(C)$ time complexity, where C is a constant. The complexity only continuously increases as a result of this algorithm. Figure 10 shows sample code of Genetic algorithms.

All experiments results are done and executed in the laptop and its specification is shown in Table 10.

```
function [child1 , child2] = crossover(parent1 , parent2, Pc, crossoverName)

switch crossoverName
    case 'single'
        Gene_no = length(parent1.Gene);
        ub = Gene_no - 1;
        lb = 1;
        Cross_P = round (  (ub - lb) *rand() + lb  );

        Part1 = parent1.Gene(1:Cross_P);
        Part2 = parent2.Gene(Cross_P + 1 : Gene_no);
        child1.Gene = [Part1, Part2];

        Part1 = parent2.Gene(1:Cross_P);
        Part2 = parent1.Gene(Cross_P + 1 : Gene_no);
        child2.Gene = [Part1, Part2];

    case 'double'
        Gene_no = length(parent1);

        ub = length(parent1.Gene) - 1;
        lb = 1;
        Cross_P1 = round (  (ub - lb) *rand() + lb  );

        Cross_P2 = Cross_P1;

        while Cross_P2 == Cross_P1
            Cross_P2 = round (  (ub - lb) *rand() + lb  );
        end

        if Cross_P1 > Cross_P2
            temp = Cross_P1;
            Cross_P1 = Cross_P2;
            Cross_P2 = temp;
        end

        Part1 = parent1.Gene(1:Cross_P1);
        Part2 = parent2.Gene(Cross_P1 + 1 :Cross_P2);
        Part3 = parent1.Gene(Cross_P2+1:end);

        child1.Gene = [Part1 , Part2 , Part3];
```

**Figure 10.** A genetic algorithm code sample.

**Table 10.** Computer specifications.

| Machine Name | Processor Type | Processor Speed | Memory | GPU | Complexity Analyzes |
|---|---|---|---|---|---|
| HP 250 G7 laptop | Intel Core i3 (7th Gen) | Speed 2.3 GHz | 8 GB | Intel HD Graphics 620 | $O(i * n2 + n * t * \log(n))$ |

## 5. Conclusions

The proposed method uses artificial neural networks (ANN) to train the network, utilizing hidden layer neurons. The training and testing costs of a prediction system for predicting the success rate of a pregnancy based on readings of the pregnancy hormone ratio in the blood were calculated using the proposed method, which combined the genetic algorithm (GA) and bat algorithm. On the other hand, one neuron in the output layer provides a pregnancy success rate. To identify the layers and nodes in the hidden layers, an iterative approach was used. A genetic, neural network, and Bat algorithm were also used to train the system. A 96.5 percent classification accuracy was achieved in simulation runs using data from the gynecological clinics at Jordan University Hospital (JUH).

**Author Contributions:** L.S. and N.A. collect data with supervision with K.F., M.A. and L.S. prepare the code while S.S. help in writing and N.S. design the methodology. All authors have read and agreed to the published version of the manuscript.

**Funding:** This research received no external funding.

**Institutional Review Board Statement:** Not applicable.

**Informed Consent Statement:** Not applicable.

**Data Availability Statement:** Data is available at the correspondence author.

**Conflicts of Interest:** The authors declared no conflict of interest.

## Nomenclature

| | |
|---|---|
| ANN | Artificial neural network |
| GA | Genetic algorithm |
| AI | Artificial intelligence |
| FSH | Follicle-stimulating hormone |
| E2 | Estrogen |
| LH | Luteinizing hormone |
| TSH | Thyroid-stimulating hormone |
| AMH | Anti-Müllerian hormone |
| FMC | First menstrual cycle |
| BESSs | Battery energy storage systems |
| RESs | Renewable energy sources |
| SOH | State of health |
| HEPC | Electric power consumption |
| JUH | Jordan University Hospital. |
| MLP | Multilayer perceptron |
| LSTM | Long short-term memory |
| LR | Linear regression |
| TAM | Technology acceptance model |
| CFS | Correlation-based feature selection method |
| RF | Random forest |
| PCA | Principal component analysis |
| IGR | Information gain ratio-based feature selection |
| ASW | Adoption of the smartwatch |
| PEU | Perceived ease of use |
| SVM | Support vector machine |
| ConvLSTM | Convolutional long short-term memory |
| RMSE | Root mean squared error |
| IHEPC | Individual household electric power consumption |
| CSV | Comma-separated Values |
| DEMS | Domestic energy management system |

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
