# Peer review of "Male and Female Hormone Reading to Predict Pregnancy Percentage Using a Deep Learning Technique: A Real Case Study"

_ai, doi:10.3390/ai3040053_

Round 1

Reviewer 1 Report

In this paper, a deep learning technique for male and female hormones reading to predict pregnancy percentage. There are a few major issues that need to be resolved before this manuscript can be accepted.

1)      The abstract should be revised as it does not enough chiefly introduce the area of research along with the research question.

2)      The language is poor and needs polishing.

3)      Research gap is not defined.

4)      Time series analysis is a challenging task. However, a discussion on it is missing in the introduction and literature section. Following is a recent paper related to time series analysis: (https://doi.org/10.3390/math9243326)

5)      The presentation of the paper is not good

6)      Motivation is not clear.

7)      I suggest the authors write their main contributions in bullets.

8)      I suggest the authors use some recent review papers to summarize the state-of-the-art, discuss the main challenges, and compared their results with the proposed approach.

9)      It is critical to point out how to set the hyper-parameters of the machine learning methods. how can we know that the tuning of the parameter will not affect the accuracy of the methods?

10)  I suggest the authors provide the complexity analysis of their proposed method. For example, model inference time on CPU and GPU and its comparison with state-of-the-arts.

Author Response

please find the response in the attached file

Reviewer 2 Report

Congratulations to the authors for proposing a method that uses Artificial Neural Networks to Predict Pregnancy.

The scientific methodology used was rigorous.

The parameters used should be increased by including factors favoring sterility such as infections, malformations, associated diseases (diabetes, etc.) and more.

Author Response

(The authors gave the same response as above.)

Reviewer 3 Report

This paper studies Male and Female Hormones Reading to Predict Pregnancy Percentage Using A Deep Learning Technique-A Real Case Study. Finally, they achieve a 96.5% classification accuracy from the experiments made on the data that were taken from 35207 patients. There are a few weaknesses that should be addressed in this paper. Therefore, I suggest the authors resubmit it after a major revision. My suggestions are as follows:

1. I strongly suggest that the paper be proofread and reread meticulously again, particularly in regard to the spelling and grammatical mistakes.

2. The paper should be revised to include at least15 recent references. You have provided only 14 references which is too low for this journal.

3. Some of your figures are pictures, and they are not visible. They should be editable.

4.   In Fig1, you provided a flowchart to explain the general framework. This section must provide a concise and clear explanation of the suggested approach. Although the flowchart is beneficial, it’s also important to outline the methodology behind this new approach.

5. In the figures you have provided a picture of the output of the suggested software which is not professional. They need more explanations.

6. The structure of your paper is weird. Please consider the MDPI structure.

7. Where is your discussion part?

8. This version is not acceptable at all and you should improve all parts with more clarification. 

Author Response

(The authors gave the same response as above.)

Reviewer 4 Report

This paper use Artificial neural networks and GA using Bat algorithm for training model accuracy checking. This paper is well presented even the novelty is poor. So, this paper may be considered after addressing the below comments.

1. There are several typos throughout the paper such as coma, semicolons, dots etc. For example, Semicolon is missing after "Artificial Neural Networks (ANN)" in the keywords.

2. The literature is not state-of-the-art. It is recommended to consider the most recent literature . It is also recommended to summarize the limitations and how this (proposed) work addressing those limitations.

3. Highlight/summarize the contributions of the paper in the introduction.

4. The discussion on machine learning testing is missing.

5. It is recommended to provide the methodology through an illustrative example for better understanding.

6. the experimental results are poor. It is recommended to extend them using multiple datasets.

7. The source code of this paper is cited in the paper using github link or personal web pages.

Author Response

(The authors gave the same response as above.)

Round 2

Reviewer 1 Report

Most of the comments and suggestions are addressed. However, the paper presentation is still not good. 

1. Contributions should be in bullet form in the introduction section.

2. Paper structure paragraph is missing, which is the last paragraph in the introduction section.

3. Each figure and table have a proper and meaningful caption.

4. Abbreviations should be defined on the first appearance and also include in table form for the reader's convenience. 

5. Each subsection should have a proper number like 4.1 or 4.1.1 etc. 

6. Table 7 is very confusing please revise it carefully. What is column sum? Furthermore, one column is not showing.

7. Please follow the following paper, it will help you to organize your research work properly.

"AB-Net: A Novel Deep Learning Assisted Framework for Renewable Energy Generation Forecasting"

Reviewer 2 Report

The new version of manuscript certainly increases the scientific value of the results presented

Reviewer 3 Report

This version is acceptable. Please add the structure of the paper at the end of the introduction. For example:

Section 2 explains the methodology and statistical methods. Section 3 demonstrates...

This can help future readers for more clarification. 
